# Real-World Data from a Refractory Triple-Negative Breast Cancer Cohort Selected Using a Clinical Data Warehouse Approach

**DOI:** 10.3390/cancers13225835

**Published:** 2021-11-21

**Authors:** Hana Kim, Hyo Jung Kim, Hongsik Kim, Hye Ryeon Kim, Hyunji Jo, Joohyun Hong, Ryul Kim, Ji-Yeon Kim, Jin Seok Ahn, Young-Hyuck Im, Se Kyung Lee, Haeyoung Kim, Soo-Yong Shin, Yeon Hee Park

**Affiliations:** 1Division of Hematology-Oncology, Department of Medicine, Samsung Medical Center, Sungkyunkwan University School of Medicine, Seoul 06351, Korea; hanafabulous@gmail.com (H.K.); hongsik.dr.kim@samsung.com (H.K.); hyeryeon90.kim@samsung.com (H.-R.K.); hyunji.jo@samsung.com (H.J.); joohyun.hong@samsung.com (J.H.); r.kim@samsung.com (R.K.); jyeon25.kim@samsung.com (J.-Y.K.); jinseok.ahn@samsung.com (J.-S.A.); yh00.im@samsung.com (Y.-H.I.); 2Center for Research Resource Standardization, Research Institution for Future Medicine, Samsung Medical Center, Seoul 06351, Korea; arcane915@gmail.com (H.-J.K.); soo-yong.shin@samsung.com (S.-Y.S.); 3Samsung Advanced Institute for Health Sciences & Technology (SAIHST), Sungkyunkwan University, Seoul 06351, Korea; 4Division of Breast Surgery, Department of Surgery, Samsung Medical Center, Sungkyunkwan University School of Medicine, Seoul 06351, Korea; sekyung.lee@samsung.com; 5Departments of Radiation Oncology, Samsung Medical Center, Sungkyunkwan University School of Medicine, Seoul 06351, Korea; haeyoung0131.kim@samsung.com; 6Department of Intelligent Precision Healthcare Convergence, Sungkyunkwan University, Suwon 16419, Korea

**Keywords:** metastatic breast cancer, triple-negative breast cancer, CDW

## Abstract

**Simple Summary:**

Patients with metastatic triple-negative breast cancer (mTNBC) have a very poor prognosis. We assume that some mTNBC patients have worse treatment outcomes and defined them as cases of refractory TNBC. We tried to investigate the characteristics and treatment outcomes of our refractory mTNBC cohort selected using a clinical data warehouse (CDW) approach. Between January 1997 and December 2019, TNBC patients were searched for in the breast cancer registry and, among them, pathologically confirmed mTNBC patients were selected as the study cohort (*n* = 451). Refractory TNBC was defined as cases of TNBC with confirmed distant metastasis within one year after adjuvant treatment. The refractory mTNBC group was younger and had a higher proportion of Ki-67 ≥ 3+ than the nonrefractory group. In addition, a much poorer prognosis existed among mTNBC patients, with an overall survival (OS) of 14.3 months and progression-free survival (PFS) of 4.2 months after first-line palliative chemotherapy compared to an OS of 24.8 months and PFS of 6.2 months in the nonrefractory TNBC group (*p* < 0.001).

**Abstract:**

Purpose: Triple-negative breast cancer (TNBC) is well known for its aggressive course and poor prognosis. In this study, we sought to investigate clinical, demographic, and pathologic characteristics and treatment outcomes of patients with refractory, metastatic TNBC selected by a clinical data warehouse (CDW) approach. Patients and methods: Data were extracted from the real-time breast cancer registry integrated into the Data Analytics and Research Window for Integrated Knowledge C (DARWIN-C), the CDW of Samsung Medical Center. Between January 1997 and December 2019, a TNBC cohort was searched for in the breast cancer registry, which includes records from more than 40,000 patients. Among them, cases of pathologically confirmed metastatic TNBC (mTNBC) were selected as the cohort group (*n* = 451). The extracted data from the registry via the CDW platform included clinical, pathological, laboratory, and chemotherapy information. Refractory TNBC was defined as confirmed distant metastasis within one year after adjuvant treatment. Results: This study comprised a total of 451 patients with mTNBC, including 69 patients with de novo mTNBC, 131 patients in the nonrefractory TNBC group with confirmed stage IV disease after one year of adjuvant treatment, and 251 patients with refractory mTNBC, whose disease recurred as stage IV within one year after completing adjuvant treatment. The refractory mTNBC cohort was composed of patients with disease that recurred at stage IV after surgery (refractory mTNBC after surgery) (*n* = 207) and patients in whom metastasis was confirmed during neoadjuvant chemotherapy (unresectable TNBC due to progression during neoadjuvant chemotherapy) (*n* = 44). Patients in the refractory mTNBC group were younger than those in the nonrefractory group (median age 46 vs. 51 years; *p* < 0.001). Considering the pathological findings, the refractory group had a greater proportion of cases with Ki-67 ≥ 3+ than did the nonrefractory group (71% vs. 47%; *p* = 0.004). During a median 8.4 years of follow-up, the overall survival was 24.8 months in the nonrefractory mTNBC group and 14.3 months in the refractory mTNBC group (*p* < 0.001), and the median progression-free survival periods were 6.2 months and 4.2 months, respectively (*p* < 0.001). The median disease-free survival period was 30.1 months in the nonrefractory mTNBC group and only 7.6 months in the refractory mTNBC group. Factors related to metastatic sites affecting overall survival were liver metastasis at diagnosis (*p* < 0.001) and leptomeningeal involvement (*p* = 0.001). Conclusions: We revealed that patients with refractory mTNBC had a much poorer prognosis among all mTNBC cases and described the characteristics of this patient group.

## 1. Introduction

Breast cancer is the most common cancer diagnosed in women worldwide. Triple-negative breast cancer (TNBC) is a type of breast cancer that does not express estrogen receptor (ER), progesterone receptor (PR), or human epidermal growth factor receptor 2 (HER2) genes [1,2]. It accounts for 10% to 15% of newly diagnosed breast cancer cases [3] and is well-known as the most aggressive subtype of breast cancer. Metastatic TNBC (mTNBC) is reported to impart a very poor prognosis, with a median overall survival (OS) period of nine to 17 months despite standard chemotherapy because it is not manageable with targeted therapies such as endocrine therapy or HER2-targeted therapies [4,5]. Recently, incorporation of immune checkpoint inhibitors has shown a significant benefit for patients with mTNBC as a first-line treatment [6,7]. There are few reports on long-term follow-up of clinical and pathological characteristics and the natural course of TNBC [8,9,10,11].

When treating mTNBC patients in a clinical setting, we have observed a certain group of patients whose disease progression worsens more rapidly than that of others. In response to this, other researchers have also attempted to identify the characteristics of patients with rapid disease progression [12,13]. We wondered which patients’ disease courses progressed faster than others. Considering this, we newly defined a certain group of patients whose disease progression worsens more rapidly than others as a group of patients who experienced disease recurrence with distant metastasis within one year after completion of adjuvant treatment (adjuvant radiotherapy or chemotherapy). We attempted to analyze how this refractory mTNBC group differs from the general, nonrefractory mTNBC group and the de novo mTNBC group, respectively.

At the beginning of our study, a clinical data warehouse (CDW) approach was chosen to extract the data of patients rather than the ordinary chart review method. Because retrospective studies have a number of limitations, such as, investigator bias, nonrepresentative patient sample, and unsystematic data collection [14], we used a CDW approach to overcome these issues of our retrospective study and to increase its objectivity and accuracy. In this way, we tried to determine how well long-term follow-up, real-world data of TNBC can be defined. It will be helpful to understand the exact clinicopathological characteristics of the refractory mTNBC population and the factors affecting patient prognosis to assist clinicians who must make decisions in a clinical setting.

## 2. Materials and Methods

### 2.1. Data Source: The SMC Breast Cancer Registry

Data were extracted from a real-time breast cancer registry (BCR, Figure 1) integrated into the Data Analytics and Research Window for Integrated Knowledge (DARWIN)-C, the clinical data warehouse of Samsung Medical Center (SMC), a big single center. Records of more than 40,000 patients with breast cancer have been collected in this registry since 1995, according to the following criteria: (i) International Classification of Diseases code-based classification of breast cancer (C50, D05), (ii) at least one visit to the clinical department of the SMC Breast Cancer Center, and iii) older than 15 years of age at enrollment. Available clinical variables in the form of structured data from the registry comprise demographics, treatment (operation, chemotherapy, radiation therapy), test results (laboratory, pathology, radiology), selected comorbidities, and targeted panel genomic test results. To ensure the real-world data reliability of the electronic health record (EHR)-based registry, a massive data quality-control effort was made concerning all core variables via descriptive statistics review by a multidisciplinary expert committee [15].

### 2.2. Study Population

The study population was derived from the SMC breast cancer registry. We identified 37,424 patients from the registry who had their first visit between January 1997 and December 2020 to a breast cancer center-related clinical department of SMC. Computed tomography (CT) scans of the chest and abdomen were performed to confirm metastatic lesions at the initial staging workup. The mTNBC cohort included women diagnosed with pathologically confirmed TNBC with a history of at least one palliative treatment for metastasis (*n* = 451). Based on the progression event time of treatment resumption in the adjuvant setting or reoperation, the mTNBC cohort was divided into two groups: patients with refractory mTNBC (*n* = 251) and those with nonrefractory TNBC (*n* = 131). The refractory TNBC group was divided into a subgroup who experienced recurrence with distant metastasis after surgery and adjuvant treatment (*n* = 207) and a subgroup with disease that progressed to stage IV during neoadjuvant chemotherapy (NAC) (*n* = 44). In addition, a group of de novo mTNBC patients (*n* = 69) was analyzed for comparison with the refractory and nonrefractory mTNBC groups. All searches in the study were conducted via the DARWIN-C platform from the EMRs of SMC, which is a large, single-center CDW; thus, we assessed only de-identified, anonymous patient data collected from the CDW.

The study protocol was approved by the Institutional Review Board of SMC Seoul, Korea (approval no. 2018-11-050-009), and the study was conducted in accordance with the ethical principles of the Declaration of Helsinki and the Korea Good Clinical Practice guidelines. All patients provided written informed consent before enrollment. Written informed consent was obtained from the patients.

### 2.3. Clinical and Pathological Definitions

The clinical staging of breast cancer was based on the American Joint Committee on Cancer’s seventh edition staging manual. TNBC was defined as an expression level less than 1% ER and PR and lack of HER2 expression during gene amplification (as defined by guidance of the American Society of Clinical Oncology and College of American Pathologists) [16,17]. Histologic grade was estimated using the Bloom–Richardson score system. The Ki-67 index is calculated by the percentage of the positive stained tumor nuclear cells out of all tumor cells. It ranged from 0% to 100% and we defined 1+ (≤25%), 2+ (25–50%), 3+ (51–75%), and 4+ (76–100%). A pathological complete response (pCR) was defined as the absence of residual invasive cancer or a hematoxylin and eosin evaluation of the complete resected breast specimen and all sampled regional lymph nodes following completion of neoadjuvant therapy [18].

### 2.4. Statistical Analysis

Data was analyzed from initial diagnosis until death, end of enrollment, or end of the observation period. Descriptive statistics are reported as proportions and medians. Data is presented as a percentage (%) for categorical variables. The Chi-square test was used to compare the distribution of categorical variables. One-way analysis of variance and the Kruskall-Wallis test were used to confirm the differences among de novo mTNBC, the nonrefractory mTNBC, and the refractory mTNBC groups. The clinicopathological differences between the nonrefractory mTNBC and refractory TNBC groups were evaluated using the Chi-square test and Fisher’s exact test.

Response evaluations were assessed according to the Response Evaluation Criteria in Solid Tumors, version 1.1 [19,20]. Disease-free survival (DFS) was defined as the duration of time from surgical resection to the date of confirmed distant recurrence. Progression-free survival was defined as the time from the first day of treatment to the date of confirmed disease progression or to the start of second line palliative chemotherapy. OS was defined as the time from the first day of first-line palliative treatment to the date of death. Kaplan–Meier estimates were used in the analysis of all time to event variables, and the 95% confidence interval for the median time to event was computed. Cox regression models were applied to assess the factors associated with OS. Statistical significance was considered at *p* < 0.05, and all its levels were two-sided. All statistical analyses were conducted using the Statistical Package for the Social Sciences version 27 (IBM Corporation, Armonk, NY, USA).

## 3. Results

### 3.1. Patient Characteristics

We started patient selection using the CDW approach from January 1997 to December 2019 at SMC. The total number of patients diagnosed with breast cancer was 37,424. Among them, selected TNBC patients totaled 3531 according to immunohistochemistry test results. Next, we searched for patients who had a history of at least one round of palliative chemotherapy session at SMC. As a result, 451 metastatic TNBC patients with a median follow-up period of 8.4 years (range, 1.4–24.8 years) were included in the analysis (Figure 2). Each patient was classified as either a case of de novo mTNBC, nonrefractory mTNBC, or refractory mTNBC. The refractory mTNBC group included patients with distant recurrence of disease within one year of completing adjuvant treatment (chemotherapy or radiotherapy). This group was analyzed by subdividing it into patients who experienced recurrence with distant metastasis within one year of completion of adjuvant treatment after surgery (refractory mTNBC with surgery subgroup) and patients with confirmed distant recurrence in the middle of NAC before surgery (unresectable mTNBC due to progression subgroup).

The median age at the time of diagnosis was 49 years in the de novo mTNBC, 51 years in the nonrefractory mTNBC group, and 46 years in the refractory mTNBC group. The proportions of patients younger than 40 years were 22%, 15%, and 33%, respectively, with the highest rate occurring in the refractory mTNBC group (Table 1). The refractory mTNBC group was statistically younger than the nonrefractory TNBC group (*p* < 0.001).

Most cases were identified as the infiltrating ductal carcinoma (IDC) histologic subtype, and there was no difference among the other groups. The ratio of Ki-67 ≥ 3+ was confirmed as 67% for the de novo mTNBC group, 47% for the nonrefractory mTNBC group, and 71% for the refractory mTNBC group. There was a significant difference between the nonrefractory mTNBC and refractory mTNBC groups regarding Ki-67 ≥ 3+ (*p* = 0.004).

The most frequently confirmed metastasis site in the de novo group at the time of diagnosis of mTNBC was bone (*n* = 29, 42%). In the nonrefractory mTNBC and refractory mTNBC groups, lung metastasis was confirmed as the most common lesion, accounting for 45% and 36% of cases, respectively. The brain metastasis rate was higher in the refractory group (*n* = 25, 14%) than in the nonrefractory group (*n* = 8, 6%) (*p* = 0.001). The additional information of patients’ characteristics is attached in Appendix A.

### 3.2. Chemotherapy

Considering those who received first-line palliative chemotherapy, the nonrefractory mTNBC group received the largest number of chemotherapy regimens in the order of taxane only (*n* = 38, 29%), taxane + platinum (*n* = 34, 26%), and capecitabine (*n* = 25, 19%). In the refractory mTNBC group, the most frequently used chemotherapy was gemcitabine + platinum (GP) (*n* = 71, 28%), followed by taxane + platinum (*n* = 63, 25%) and taxane only (*n* = 49, 20%). In the refractory mTNBC after surgery subgroup, the same order was observed. Second-line palliative chemotherapy was performed at similar rates of 83% in the de novo, 80% in the nonrefractory, and 80% in the refractory mTNBC group. The nonrefractory mTNBC group received second-line chemotherapy in the order of capecitabine (*n* = 39, 37%) and GP (*n* = 27, 26%), whereas the refractory mTNBC group was treated with chemotherapy in the order of GP (*n* = 51, 25%) and capecitabine (*n* = 45, 22%).

In the nonrefractory mTNBC group, 24% of patients received NAC, and a pCR was confirmed in 6% of them. Meanwhile, in the refractory mTNBC group, 63% of patients received NAC, and a pCR was confirmed in 3% (Table 2). Concerning the histologic grade confirmed in surgical specimens, the refractory group was confirmed to have a higher proportion (*n* = 156, 75%) of poorly differentiated grades than the nonrefractory mTNBC group (*n* = 76, 58%). There was no significant difference in pathologic tumor and node staging between the two groups. The percentage of leptomeningeal involvement confirmed after stage IV diagnosis was 10% (*n* = 21) in the refractory mTNBC group, which was higher than that of 4% (*n* = 5) in the nonrefractory mTNBC group (*p* = 0.009).

### 3.3. Survival and Prognosis

The median DFS from surgery in the nonrefractory group was 30.1 months, and that in the refractory mTNBC group was 7.6 months (Table 3). The PFS after first-line chemotherapy was 3.8 months in the de novo group, 6.2 months in the nonrefractory group, and 4.2 months in the refractory mTNBC group. The OS was 17.3 months in the de novo group, 24.8 months in the nonrefractory group, and 14.3 months in the refractory mTNBC group. The OS of the refractory mTNBC with surgery subgroup was 14.4 months. The differences in the OS (*p* < 0.001) and PFS (*p* < 0.001) among groups were significant (Figure 3).

Factors affecting OS were analyzed by Cox regression analysis. Bone metastasis (*p* = 0.018), liver metastasis (*p* < 0.001), and leptomeningeal involvement (*p* = 0.002) were identified as significant factors in univariable analysis. As a result of multivariable analysis, liver metastasis (*p* < 0.001) and leptomeningeal seeding (*p* = 0.002) were confirmed as significant (Table 4).

## 4. Discussion

In this study, we newly defined a refractory mTNBC group and selected eligible patients, extracted their data, and analyzed real-world data of this group from the registry via the CDW platform. Electronic health record (EHR) data as a type of real-world data has the potential to further the research process from expert inference to the creation of real-world evidence due to its fine information granularity and selection bias mitigation. The CDW is a platform used to integrate multiple data sources digitalized via the EHR, including data processing and analysis tools [21]. Although the number of institutions implementing in-house CDW is increasing, transforming EHR data into analyzable data with enhancing completeness, and the level of content reliability required for research in the context of secondary use of clinical data is challenging [22]. Using real-world data, we can learn about patients who cannot be enrolled in clinical trials, such as patients with refractory TNBC who are most likely to require aggressive treatment. Actually, most clinical trials for TNBC patients have been limited to only nonrefractory TNBC patients. Only one clinical trial has included refractory TNBC patients (Impassion132, NCT03371017). In addition, concerns remain regarding use of real-world data for cancer research, such as how to control the complexity or robust methods to control variability of extracted research variables by independent researchers, ensuring data quality from the primary datasets [23,24]. To address these pitfalls, the SMC breast cancer registry integrated into a CDW was constructed in 2016. Disease-specific core variables were declared by clinicians, and the extract transform load (ETL) pipeline from raw EHR data to featured variables was implemented in the CDW platform for transparency and consistency of the data-transformation process.

A characteristic feature of the metastatic TNBC (both refractory and nonrefractory) patient group in this study was the very low pCR rates, i.e., 6% in the nonrefractory mTNBC group and 3% in the refractory mTNBC group. pCR rates of approximately 30% to 40% have been achieved for patients with TNBC with chemotherapy alone [18,25,26]. Considering that the pCR rate is very low in patients with mTNBC, it can be inferred that distant recurrence might not occur if pCR is achieved after NAC. Therefore, it is important to increase the pCR rate after NAC to improve the prognosis of TNBC patients. Recently, various attempts have been made to increase the pCR rate by adding immune checkpoint inhibitors to NAC [7,27]. In this way, increasing the pCR rate by trying new treatment drugs might be a solution for improving the prognosis of TNBC patients.

In this study, the refractory TNBC group had a worse prognosis (median OS of 14.4 months) compared to reports from recent phase III clinical trials. In the IMpassion130 trial, the median OS was 21.3 months in the atezolizumab-nab-paclitaxel group and 17.6 months in the placebo-nab-paclitaxel group [6]. Another phase III clinical trial, IMpassion131, documented a median OS of about 20 months in patients receiving paclitaxel with or without aztezolizumab [28]. This might be because these clinical trials did not include TNBC patients who might have experienced worse treatment outcomes. Therefore, real-world data such as in our study will be helpful in understanding the real clinical aspects of refractory TNBC patients.

Our study had several limitations. This study was a retrospective analysis, and the study cohort was derived from a single large center in Korea. As a tertiary hospital, our patient cohort might not represent other institutions’ metastatic TNBC patients. In addition, our patient population was of a single ethnicity, and its analysis might have yielded results different from those of other ethnic groups. Brain imaging was not performed at regular intervals, but performed following a clinician’s decision while the symptoms of brain metastasis occurred. Hence the brain metastasis might differ from the actual occurrence of brain metastasis. In addition, the defined PFS period for data extraction, from the first line chemotherapy until the start of second line chemotherapy is generally consistent but might differ from the correct PFS period. We could not analyze all factors that might affect the patients’ prognosis, such as clinical stages, comorbidities, and other blood lab results. It is possible that these variables could be confounding factors.

However, in this study, we performed patient selection and data collection using a real-time site-specific cancer registry integrated into a CDW approach, and these efforts could have reduced human errors. Furthermore, the research approach suggested by the study using a preprocessed dataset with data quality measures can help accelerate the generation of real-world evidence by facilitating the cycle of hypothesis derivation and verification. Another limitation is that patients with a history of systemic chemotherapy were selected during the development of the initial study cohort. Therefore, it is possible that patients who only received local treatment such as intrathecal methotrexate or whole-brain radiation therapy but not systemic chemotherapy were excluded from the baseline cohort. Patients who did not receive systemic chemotherapy due to poor general condition were also not included in this cohort. For this reason, it is estimated that the distant recurrence rate of 12.8% (451 of 3531) of TNBC shown in this paper seemed to be lower than the distant recurrence rates of around 20% presented in other articles [29,30,31].

In this study, the poor prognosis of mTNBC patients was reconfirmed, and, in particular, the refractory mTNBC group experienced shorter OS despite being composed of younger patients and receiving more aggressive chemotherapies. The refractory mTNBC with surgery subgroup had a poor prognosis like the refractory mTNBC group diagnosed with stage IV disease during NAC. Unfortunately, refractory mTNBC patients are a rapidly progressing group of patients who do not meet the inclusion criteria of many clinical trials, which limits their access to new treatments. More clinical trials should be performed to explore new treatment options that can improve the prognosis of refractory mTNBC patients.

## 5. Conclusions

In this study, we defined newly defined a refractory mTNBC group and selected eligible patients, extracted their data, and analyzed real-world data of this group from the registry via the CDW platform. The refractory mTNBC group was younger and had a higher proportion of Ki-67 ≥ 3+ than the nonrefractory group. Moroever, a much poorer prognosis existed among mTNBC patients, with an overall survival (OS) of 14.3 months and progression-free survival (PFS) of 4.2 months after first-line palliative chemotherapy compared to an OS of 24.8 months and PFS of 6.2 months in the nonrefractory TNBC group (*p* < 0.001). This refractory mTNBC patients had poor prognosis and survival outcome, so we should consider more aggressive treatment strategy such as clinical trials in clinical setting.

## Figures and Tables

**Figure 1 cancers-13-05835-f001:**
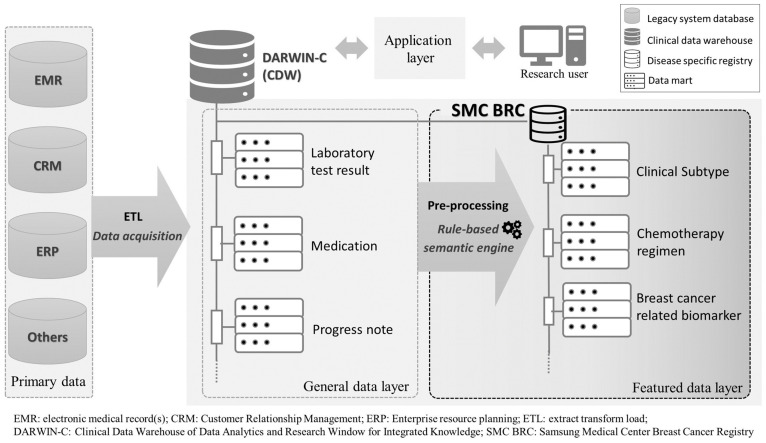
The Samsung Medical Center breast cancer registry (SMC BCR); clinical data warehouse integrated data architecture.

**Figure 2 cancers-13-05835-f002:**
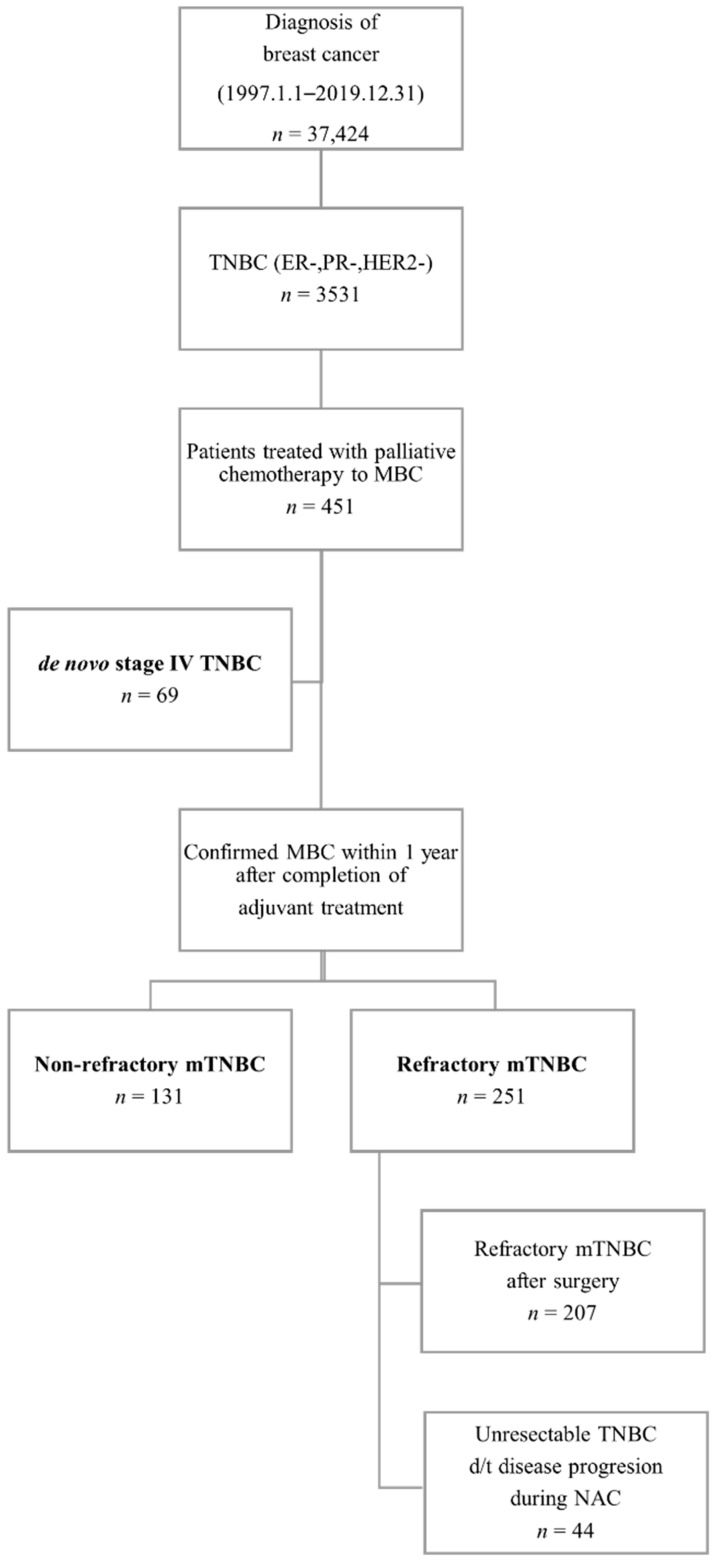
A flowchart of patient selection via a clinical data warehouse approach. Abbreviations: TNBC, triple-negative breast cancer; ER, estrogen receptor; PR, progesterone receptor; HER2, human epidermal growth factor receptor 2; MBC, metastatic breast cancer.

**Figure 3 cancers-13-05835-f003:**
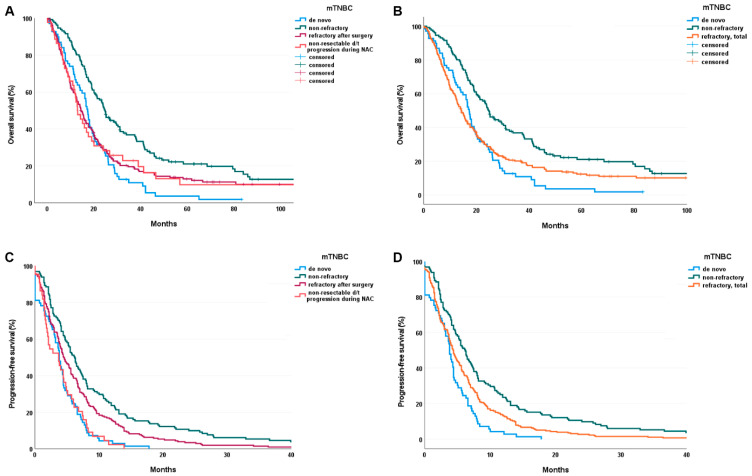
(**A**,**B**) Overall survival of refractory and total (refractory after surgery + non-resectable TNBC d/t progression) groups (*p* < 0.001). (**C**,**D**) Progression-free survival of refractory and total (refractory after surgery + non-resectable TNBC d/t progression) groups (*p* < 0.001). Abbreviation: mTNBC, metastatic triple-negative breast cancer.

**Table 1 cancers-13-05835-t001:** Characteristics differences between de novo, nonrefractory metastatic triple-negative, and refractory metastatic triple-negative breast cancer after surgery and unresectable triple-negative breast cancer due to progression during neoadjuvant chemotherapy.

Group	De Novo mTNBC	Recurrence of mTNBC after Curative Surgery		
		Nonrefractory mTNBC	Refractory mTNBC	
			Refractory mTNBC after Surgery	Unresectable mTNBC d/t Progression during NAC
Patient, no.	69	131	207	44
			251	
Age				
At diagnosis, median, yr. (range)	49 (28–89)	51 (29–80)	45 (24–90)	46 (28–76)
			46 (24–90)	
<40	15 (22)	20 (15)	67 (32)	16 (36)
			83 (33)	
<60	53 (77)	111 (85)	192 (93)	41(93)
			233 (93)	
≥60	16 (23)	20 (15)	15(7)	3 (7)
			18 (7)	
Menopausal status at diagnosis				
Premenopause	29 (42)	52 (40)	130 (63)	27 (61)
			157 (63)	
Postmenopause	34 (49)	64 (49)	64 (31)	14 (32)
			78 (31)	
Unknown	6 (9)	15 (11)	13 (6)	3 (7)
			16 (6)	
Histologic subtypes				
Ductal	62 (90)	113 (86)	180 (87)	43 (98)
			223 (89)	
Lobular	1 (1)	3 (2)	2 (1)	0
			2 (1)	
Metaplastic	1 (1)	9 (7)	17 (8)	0
			17 (7)	
Other	5 (7)	6 (5)	8 (4)	1 (2)
			9 (4)	
Ki-67				
1+	6 (9)	29 (22)	19 (9)	6 (14)
			25 (10)	
2+	12 (17)	25 (19)	38 (18)	7 (16)
			45 (18)	
≥3	46 (67)	62 (47)	146 (71)	31 (70)
			177 (71)	
Unknown	5 (7)	15 (11)	3 (1)	0
			3 (1)	
*BRCA* mutation				
*BRCA1*	3/13 (23)	2/15 (13)	9/53 (17)	1/5 (20)
			10/58 (17)	
*BRCA2*	1/13 (8)	0/15	3/53 (6)	0/5
			3/58 (5)	
Distant metastasis site at diagnosis of MBC				
Bone	29 (42)	36 (27)	51 (25)	11 (25)
			62 (25)	
Brain	0	8 (6)	31 (15)	4 (9)
			35 (14)	
Liver	16 (23)	20(15)	34 (16)	6 (14)
			40 (16)	
Lung	24 (35)	59 (45)	73 (35)	18 (41)
			91 (36)	
First-line palliative chemotherapy				
AC	35 (51)	5 (4)	7 (3)	4 (9)
Taxane only	9 (13)	38 (29)	43 (21)	6 (14)
			49 (20)	
Taxane + G	0	15 (11)	8 (4)	1 (2)
Taxane + platinum	10 (15)	34 (26)	51 (25)	13 (30)
			64 (25)	
GP (C)	3 (4)	6 (5)	60 (29)	11 (25)
			71 (28)	
NX	0	0	2 (1)	1 (2)
Capecitabine	8 (12)	25 (19)	18 (9)	7 (16)
Other	4 (6)	8 (6)	18 (9)	1 (2)
Second-line palliative chemotherapy				
Patients, no.	57/69 (83)	105/131 (80)	167/207 (81)	35/44 (80)
			202 (80)	
AC	16 (28)	10 (10)	8 (5)	1 (3)
Taxane only	12 (21)	7 (7)	24 (14)	3 (9)
Taxane + G	0	1 (1)	0	0
Taxane + platinum	11 (19)	12 (11)	19 (11)	2 (6)
GP (C)	5 (9)	27 (26)	43 (26)	8 (23)
			51 (25)	
NX	1 (2)	0	4 (2)	1 (3)
Capecitabine	10 (18)	39 (37)	32 (19)	13 (37)
			45 (22)	
Other	2 (4)	9 (9)	37 (22)	7 (20)

Abbreviations: mTNBC, metastatic triple-negative breast cancer; AC, adriamycin/cyclophosphamide; G, gemcitabine; GP (C), gemcitabine/cisplatin (carboplatin); NX, navelbin/xeloda.

**Table 2 cancers-13-05835-t002:** Clinicopathological differences between patients with nonrefractory triple-negative breast cancer and refractory triple-negative breast cancer who could receive surgery with a curative aim.

No., (%)	Nonrefractory mTNBC	Refractory mTNBC
Patient, no.	131	207
Neoadjuvant chemotherapy		
Yes	31 (24)	130 (63)
pCR	2/31 (6)	4/130 (3)
Grade, histologic		
Well	5 (4)	4 (2)
Moderate	40 (31)	38 (18)
Poorly differentiated	76 (58)	156 (75)
Unknown	10 (8)	9 (4)
pT stage		
No residual tumor	2 (2)	4 (2)
1	46 (35)	55 (27)
2	75 (57)	101 (49)
3	8 (6)	44 (21)
4	0	3 (1)
pN stage		
0	58 (44)	83 (40)
1	41 (31)	65 (31)
2	21 (16)	30 (14)
3	11 (8)	29 (14)

Abbreviations: mTNBC, metastatic triple-negative breast cancer; pCR, pathologic complete response; MBC, metastatic breast cancer; pT stage, pathologic tumor stage; pN stage, pathologic nodal stage.

**Table 3 cancers-13-05835-t003:** Disease-free survival, progression-free survival, and overall survival of each metastatic triple-negative breast cancer group.

Group		Recurrence of mTNBC after Curative Surgery		
	De Novo mTNBC	Nonrefractory mTNBC	Refractory mTNBC	
Median, Months			Refractory mTNBC after Surgery	Nonresectable TNBC d/t Progression during NAC
DFS (range)	-	30.1 (18.3–217.9)	7.6 (7.4–17.9)	
PFS (95% CI)	3.8 (3.3–4.3)	6.2 (5.2–7.2)	4.5 (3.7–5.3)	3.7 (1.6–5.8)
			4.2 (1.6–5.8)	
2nd PFS (range)	3.7	5.3	3.0	2.1
			2.6 (0–25.0)	
OS (95% CI)	17.3 (15.9–18.7)	24.8 (21.4–28.2)	14.4 (12.5–16.5)	12.9 (9.7–16.1)
			14.3 (12.5–16.1)	

Abbreviations: mTNBC, metastatic triple-negative breast cancer; NAC, neoadjuvant chemotherapy; DFS, disease-free survival; PFS, progression-free survival; OS, overall survival; 2nd PFS, progression-free survival from second-line palliative chemotherapy to disease progression or death.

**Table 4 cancers-13-05835-t004:** Univariable and multivariable analyses of risk factors of overall survival.

Characteristics	Univariable Analysis		Multivariable Analysis	
	HR (95% CI)	*p*-Value	HR (95% CI)	*p*-Value
Age < 40	0.997 (0.793–1.252)	0.977	NA	NA
Non-IDC histology	0.677 (0.403–1.139)	0.141	NA	NA
Ki-67 4+ (76–100%)	0.978 (0.795–1.201)	0.829	NA	NA
Bone metastasis	1.299 (1.045–1.615)	0.018	1.184 (0.649–1.77)	0.135
Brain metastasis	1.162 (0.820–1.646)	0.398	NA	NA
Liver metastasis	1.947 (1.506–2.518)	<0.001	2.009 (1.552–2.600)	<0.001
Lung metastasis	0.918 (0.746–1.129)	0.416	NA	
Leptomeningeal seeding	1.744 (1.230–2.473)	0.002	1.862 (1.311–2.645)	0.001

Abbreviations: OS, overall survival; HR, hazard ratio; CI, confidence interval; IDC, infiltrating ductal carcinoma; NA, not available.

## Data Availability

The data presented in this study are available on request from the corresponding author.

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
