# Peer review of "Real-World Data from a Refractory Triple-Negative Breast Cancer Cohort Selected Using a Clinical Data Warehouse Approach"

_cancers, 2021, doi:10.3390/cancers13225835_

Round 1

Reviewer 1 Report

Dear Editor, Thanks for the opportunity to review the manuscript titled “Real-world Data from a Refractory Triple-negative Breast Can- 2 cer Cohort Selected Using a Clinical Data Warehouse Approach” by Hana Kim and coll.

The manuscript addresses an issue that is relevant, to better characterized patients with a TNBC refractory to treatment. They used a clinical data warehouse approach, an approach that have a lot of advantages. However, the objectives are not clear, and I think that authors can not conclude on their objectives with their data. There are some serious issues that the authors need to address before the manuscript can be considered for publication, especially regarding the important biases that are not discussed.

The following are my comments describing these issues.

Introduction

P2-L74 – Please add reference

P2L77 – What did they conclude? Which factors are associates with rapid disease progression in other studies?

P2L85-93 : It is unclear which method authors finally used. The paragraph begin with “at the beginning of our study”.

Introduction should end with principal and secondary objectives of the study. In my opinion, objectives are unclear.

Materials and methods

P3L101 – How many centers are included in the SMC Breast Cancer Center? Is it only one hospital, or a group of several hospitals?

P3L103-105: “e demographics, treatment (oper- ation, chemotherapy, radiation therapy), test results (laboratory, pathology, radiology), selected comorbidities, and targeted panel genomic test results” : Why all these caracteristics are not available in this study ?  It would be very important to characterize the disease and the patient.

P3L111-113: How do you identify TNBC patients before the characterization of HER2 status? When was routinely done HER2 IHC in Korea?

P3 L118 : Which imagery were performed at diagnosis ? How authors could be sure that the subgroup of patients who progressed during NAC were not de novo MBC ? I imagine that imagery evolved during the very long period of time : which imagery for which stade? Brain imagery ? We need further detail regarding the initial stadification.

P3L133 : authors included patients far before this pathological definition. How TNBC disease was defined?

P3L136 : please add a ref for pCR.

P4L148 : Is it a retrospective evaluation, or a routinely used evaluation ?

P4L149 : Do authors have information regarding locoregional reccurence or distant reccurence?

P4L151: date of disease progression or start of second line chemotherapy are very different endpoint. They should not be included in the same endpoint.

Results

Patient characteristics

P4L165: Do you have information regarding patients who relapse but who did not receive any treatment?

P4L166 : I am surprised by the low number of patients (from 3500 to 451). What about the others? No relapse, death, or loss of follow-up ?

P4L173 : for me, it seems very difficult to be sure that they were not de novo MBC patients.

P4 L189: How was realized brain evaluation at diagnosis ? systematically ? regarding stage? TDM or RMI?

Chemotherapy

P8L215 – Maybe I(ve missed the information but what was the regimen of NAC ?

P9L221 : How was defined leptomeningeal involvement?

Survival and prognosis

P9L229-233 : I am not sure that DFS and PFS are very pertinent endpoint in this study, as group of patients are defined regarding the period of relapse. By definition, DFS and PFS are different.

The important question is : which clinic-biolgical or pathological / or imagery / baseline factors could help clinicians to predict refractory TNBC patients?

P10-L240 : Refractory / nonrefractory / de novo status were not associated with OS ?

Discussion

P11L254 – It is well known in the literature that patients who have an early relapse of TNBC have a worse prognosis …. Question is : how to identify these patients at diagnosis ?

P11L271 – It means patients from this study were included from 2016? It is difficult to follow.

P11 L276 – We don’t have any information regarding the initial TNM. This could strongly impact pCR. Informations regarding the NAC regimen? How many cycles? Which type of surgery ?

There are a lot of missing information to conclude.

P12L286 – I don’t understand what authors compared. Refractory TNBC patients with patients from clinical trials? This is not the same population.

P12L295 – A huge quantity of information are missing, and they have to be discussed in this section. Clinical parameters (PS, co-morbidities, …), stage of disease, imagery at baseline, …. All these missing data obviously biased conclusions.

Figure 2 : Maybe authors could change “confirmed MBC within within 1 year … YES and NO” with

“confirmed MBC after completion of adjuvant treatment” and after 2 parts : within 1 year or after 1 year. Do we have informations regarding the 3000 other patients?

I am surprise of the low proportion of dnMBC TNBC (less than 2%). Any explanations?

Author Response

Thank you for your valuable comments. 

Reviewer 1

Dear Editor, Thanks for the opportunity to review the manuscript titled “Real-world Data from a Refractory Triple-negative Breast Can- 2 cer Cohort Selected Using a Clinical Data Warehouse Approach” by Hana Kim and coll.

The manuscript addresses an issue that is relevant, to better characterized patients with a TNBC refractory to treatment. They used a clinical data warehouse approach, an approach that have a lot of advantages. However, the objectives are not clear, and I think that authors can not conclude on their objectives with their data. There are some serious issues that the authors need to address before the manuscript can be considered for publication, especially regarding the important biases that are not discussed.

The following are my comments describing these issues.

Introduction

P2-L74 – Please add reference

Thank you for your comment. There were few papers that dealt with the clinicopathological characteristics and outcomes of metastatic TNBC in a long-term, like ours. We have attached the references of the papers with similar content below.

  1. James, M.; Dixit, A.; Robinson, B.; Frampton, C.; Davey, V. Outcomes for Patients with Non-metastatic Triple-negative Breast Cancer in New Zealand. Clin Oncol (R Coll Radiol) 2019, 31, 17-24, doi:10.1016/j.clon.2018.09.006.
  2. Polley, M.C.; Dickler, M.N.; Sinnwell, J.; Tenner, K.; de la Haba, J.; oibl, S.; Goetz, M.P.; Bergh, J.; Roberston, J.; Couch, F.; et al. A clinical calculator to predict disease outcomes in women with hormone receptor-positive advanced breast cancer treated with first-line endocrine therapy. Breast Cancer Res Treat 2021, 189, 15-23, doi:10.1007/s10549-021-06319-z.
  3. Chen, S.S.; Tang, S.C.; Li, K.; Wu, J.; Li, X.; Ren, H.; Sun, X. Predicting the Survival of Triple-Negative Breast Cancer in Different Stages: A SEER Population Based Research Referring to Clinicopathological Factors. Cancer Invest 2020, 38, 549-558, doi:10.1080/07357907.2020.1831010.
  4. Li, C.Y.; Zhang, S.; Zhang, X.B.; Wang, P.; Hou, G.F.; Zhang, J. Clinicopathological and prognostic characteristics of triple- negative breast cancer (TNBC) in Chinese patients: a retrospective study. Asian Pac J Cancer Prev 2013, 14, 3779-3784, doi:10.7314/apjcp.2013.14.6.3779.

P2L77 – What did they conclude? Which factors are associates with rapid disease progression in other studies?

  • Thank you for your comment. It seems that we had a problem with the expression when writing the part. Although metastatic TNBC itself has a poor prognosis, some patients have a worse prognosis due to faster disease progression. What we meant to say in the introduction was that we wanted know the causes of the patients getting worse so quickly. The paragraph has been amended as follows because in may have trouble understanding the content.

When treating mTNBC patients in clinical setting, we have observed a certain group of patients whose disease progression worsens more rapidly than that of others. In response to this, other researchers have attempted to identify the characteristics of patients with rapid disease progression like us. We wondered which patients’ disease courses progressed faster than others

P2L85-93 : It is unclear which method authors finally used. The paragraph begin with “at the beginning of our study”.

Introduction should end with principal and secondary objectives of the study. In my opinion, objectives are unclear.

 Thank you for your comment. We chose the clinical data warehouse (CDW) approach rather than the ordinary chart review method to overcome the limitations of the retrospective studies. Perhaps there was a problem with the expression and the content was not delivered well. So the part was changed as follows.

At the beginning of our study, a clinical data warehouse (CDW) approach was chosen to extract the data of patients rather than the ordinary chart review method.

Materials and methods

P3L101 – How many centers are included in the SMC Breast Cancer Center?

Thank you for your comment. In order to avoid confusion among readers, the sentence has been modified as follows.

Data were extracted from a real-time breast cancer registry (BCR) integrated into the Data Analytics and Research Window for Integrated Knowledge (DARWIN)-C, the clinical data warehouse of Samsung Medical Center (SMC), a large single center.

P3L103-105: “e demographics, treatment (oper- ation, chemotherapy, radiation therapy), test results (laboratory, pathology, radiology), selected comorbidities, and targeted panel genomic test results” : Why all these caracteristics are not available in this study ?  It would be very important to characterize the disease and the patient.

Thank you for your comment. Our CDW platform contains all of the data mentioned above, but not all of them are covered in this paper. We attached additional patients’ clinical stages, operation,  adjuvant RT status, and adjuvant chemotherapy regimen as a supplement table. The preparation time is only 10 days, so it may not be enough, but we did our best. As you pointed out, the data of patients’ characteristics in supplement 1 seems to be important for predicting the patient’s prognosis.

Supplement.

Supplement 1. Differences of clinical tumor, nodal stages between de novo, nonrefractory metastatic triple-negative, and refractory metastatic triple-negative breast cancer after surgery and unresectable triple-negative breast cancer due to progression during neoadjuvant chemotherapy.

10

Recurrence of mTNBC

after curative surgery

De novo mTNBC

Nonrefractory mTNBC

     Refractory mTNBC

Refractory mTNBC

after surgery

Unresectable mTNBC

d/t progression

during NAC

Patient, no.

69

131

207

44­­

cT stage

T1

T2

T3

T4

Unknown

4

13

17

24

11

36

76

14

1

4

23

119

55

9

1

4

17

19

4

0

cN stage

N0

5

55

56

            4

N1

3

33

46

            17

N2

20

25

62

            19

N3

35

16

42

            4

Unknown

6

2

1

            0

Abbreviations: mTNBC, metastatic triple-negative breast cancer

Supplement 2. Characteristic differences between patients with nonrefractory triple-negative breast cancer and refractory triple-negative breast cancer who could receive surgery with a curative aim.

No., (%)

Nonrefractory mTNBC

Refractory mTNBC

Patient, no.

131

207

Surgery type

BCS

55

80

Mastectomy

76

127

Adjuvant RT

Yes

89

156

No

42

51

Adjuvant chemotherapy

Yes

101

102

No

30

105

Adjuvant chemotherapy regimen

AC

10

9

AC-T

48

                  30

Capecitabine

1

17

CMF

13

9

FAC

21

19

T

3

8

Ohers

5

10

Abbreviations: mTNBC, metastatic triple-negative breast cancer; BCS, breast-conserving surgery; RT, radiation therapy; AC, Adriamycin/cyclophosphamide, AC-T, AC followed by paclitaxel or docetaxel; CMF, cyclophosphamide/methotrexate/fluorouracil; FAC, fluorouracil/Adriamycin/cyclophosphamide; T, paclitaxel or docetaxel.

. P3L111-113: How do you identify TNBC patients before the characterization of HER2 status? When was routinely done HER2 IHC in Korea?

Thank you for your comment. In Korea, HER2 status was routinely examined when biopsy was done before surgery. So we could characterize the HER2 status before the pathological report came out.

P3 L118 : Which imagery were performed at diagnosis ? How authors could be sure that the subgroup of patients who progressed during NAC were not de novo MBC ? I imagine that imagery evolved during the very long period of time : which imagery for which stade? Brain imagery ? We need further detail regarding the initial stadification.

Thank you for your comment. We did Chest CT, Abdomen-pelvis CT scan besides Breast US, mammography, breast MRI as an initial imaging work up in SMC. Thanks for pointing out the matter. We added the comment as follows.

Computed tomography (CT) scans of the chest and abdomen were performed to confirm metastatic lesions at the initial staging workup.  P3L133 : authors included patients far before this pathological definition. How TNBC disease was defined?Thank you for your comment. We just extracted a group of patients from a big data warehouse called CDW, and the pathologic status of the patient cohort met the criteria of TNBC.

P3L136 : please add a ref for pCR.

Thank you for your comment. We added the reference of the definition of pCR, von Minckwitz et al., 2012

  1. von Minckwitz, G.; Untch, M.; Blohmer, J.U.; Costa, S.D.; Eidtmann, H.; Fasching, P.A.; Gerber, B.; Eiermann, W.; Hilfrich, J.; Huober, J.; et al. Definition and impact of pathologic complete response on prognosis after neoadjuvant chemotherapy in various intrinsic breast cancer subtypes. J Clin Oncol 2012, 30, 1796-1804, doi:10.1200/JCO.2011.38.8595.

P4L148 : Is it a retrospective evaluation, or a routinely used evaluation ?

Thank you for your comment. It’s not retrospective evaluation but was the evaluation result at that time of treating patients.

P4L149 : Do authors have information regarding locoregional reccurence or distant reccurence?

Thank you for your comment. We studied only stage IV patients with distant recurrence in this paper.

P4L151: date of disease progression or start of second line chemotherapy are very different endpoint. They should not be included in the same endpoint.

Thank you for your comment. You’re right. Because even if disease progression does not occur, chemotherapy changed due to side effects. Conversely, even with disease progression, the general condition is not good, so chemotherapy may be delayed. However, such cases are seemed to be very few and will not significantly affect the analysis results.

Results

Patient characteristics

P4L165: Do you have information regarding patients who relapse but who did not receive any treatment?

Thank you for your comment. As you pointed out, patients who relapsed but did not receive treatment were excluded from the 451 cohort. This is because only patients who had received palliative chemotherapy were included in the first place. Therefore, it is difficult to know about patients who have relapsed but did not come to our hospital.

P4L166 : I am surprised by the low number of patients (from 3500 to 451). What about the others? No relapse, death, or loss of follow-up ?

Thank you for your comment. In this paper, it can be confirmed that the distant metastasis rate of TNBC is lower than commonly known in literature. According to published articles, the distant recurrence rate of TNBC is around 20%. It is clear that the recurrence rate is higher than our distant recurrence rate 12.9% (451 of 3500). This is because there were patients who relapsed but did not visit our center and cases where only local treatment was performed for distant recurrence. In addition, among patients with confirmed distant recurrence, patients who did not receive palliative chemotherapy due to poor general condition may also be excluded from the patient group. We described this in the discussion as follows.

Therefore, it is possible that patients who only received local treatment such as intrathecal methotrexate or whole-brain radiation therapy but not systemic chemotherapy were excluded from the baseline cohort. For this reason, it is estimated that the distant recurrence rate of 12.8% (451 of 3531) of TNBC shown in this paper was seemed to be lower than the distant recurrence rates of around 20% presented in other articles.

P4L173 : for me, it seems very difficult to be sure that they were not de novo MBC patients.

Thank you for your comment. You’ re right. At the time of diagnosis, it was confirmed that there was no distant metastasis on imaging evaluation, but it is possible that there was a metastatic lesion that was not confirmed on CT scans.

P4 L189: How was realized brain evaluation at diagnosis ? systematically ? regarding stage? TDM or RMI?

Thank you for your comment. Most of the patients who were diagnosed brain metastasis during the follow up period had symptoms according to the brain metastasis. Therefore, brain MRI scans were performed at the discretion of the clinician.

P8L215 – Maybe I(ve missed the information but what was the regimen of NAC ?

Thank you for your comment. The NAC regimen was selected by the clinician’s discretion. The most commonly used one is AC-T (, AC followed by paclitaxel or docetaxel).

P9L221 : How was defined leptomeningeal involvement?

Leptomeningeal involvement was defined as a case in which CSF cytology was positive. And If CSF cytology was negative, it was defined that there was the evidence of meningeal seeding on brain MRI and abnormal findings such as increased protein level, WBC count in CSF analysis.

Survival and prognosis

P9L229-233 : I am not sure that DFS and PFS are very pertinent endpoint in this study, as group of patients are defined regarding the period of relapse. By definition, DFS and PFS are different.??

The important question is : which clinic-biolgical or pathological / or imagery / baseline factors could help clinicians to predict refractory TNBC patients?

Thank you for your comment. If the patients recurred with distant metastasis within 1year after adjuvant treatment, these patients could be expected to have very poor prognosis. Especially if the patient is less than 40 years old or has high Ki-67 score.

P10-L240 : Refractory / nonrefractory / de novo status were not associated with OS ?

Thank you for your comment. The OS curve for each group is shown in Fig. 3, and there is a statistically significant differences in OS between non-refractory and refractory mTNBC group (P < 0.001).  

Discussion

P11L254 – It is well known in the literature that patients who have an early relapse of TNBC have a worse prognosis …. Question is : how to identify these patients at diagnosis ?

Thank you for your comment. We conclude that younger women under 40years of age and high Ki-67 % are more likely to be in the refractory TNBC.

P11L271 – It means patients from this study were included from 2016? It is difficult to follow.

Thank you for your comment. Sorry. We didn’t express the sentences well. The patients data warehouse was built in 2016. Since it also contains data on past patients before 2016. So, we were able to obtain data from patients from 1997.

 P11 L276 – We don’t have any information regarding the initial TNM. This could strongly impact pCR. Informations regarding the NAC regimen? How many cycles? Which type of surgery ?

There are a lot of missing information to conclude.

Thank you for your comment. The supplement table was added as previously mentioned.

P12L286 – I don’t understand what authors compared. Refractory TNBC patients with patients from clinical trials? This is not the same population. ???

Thank you for your comment. We wanted to emphasize that refractory TNBC patients with such poor prognosis should have a chance to get more aggressive treatment such as clinical trials.

P12L295 – A huge quantity of information are missing, and they have to be discussed in this section. Clinical parameters (PS, co-morbidities, …), stage of disease, imagery at baseline, …. All these missing data obviously biased conclusions.

 Thank you for your comment. It was not possible to include all necessary clinical parameters in this paper.

Figure 2 : Maybe authors could change “confirmed MBC within within 1 year … YES and NO” with

“confirmed MBC after completion of adjuvant treatment” and after 2 parts : within 1 year or after 1 year. Do we have informations regarding the 3000 other patients?

Thank you for your comment. Since we have a datawarehouse, we can extract the data of remaining 3080 patients except in cases where the patient followed up loss.

I am surprise of the low proportion of dnMBC TNBC (less than 2%). Any explanations?

Thank you for your comment. Those who did not receive palliative chemotherapy were not included the de novo mTNBC. Some patients only received local treatment for distant recurrence.

Reviewer 2 Report

The manuscript submitted by Kim et al. analyzed the clinicopathological characteristics and treatment outcomes of patients diagnosed with refractory metastatic triple negative breast cancer (mTNBC) in relation to a cohort of patients diagnosed with metastatic triple-negative breast cancer which responded to adjuvant treatment. To this end, the authors conducted a retrospective observational study using a clinical data warehouse (CDW) approach, comprising 415 patients which presented with mTNBC between January 1997 and December 2019 in a single institution, which were divided into refractory and non-refractory cohorts based on their response to neoadjuvant/adjuvant chemotherapy and compared in terms of clinical data, pathological data, laboratory data and treatment regimens, as well as disease-free survival and overall survival. The authors concluded that patients with refractory mTNBC were younger, had more aggressive tumor biology and significantly poorer outcomes in terms of overall survival, disease-free survival and progression-free survival compared to the non-refractory cohort. The main strength of this study is that it addresses a relevant research question, with significant implications for clinical practice.

Title and abstract: The title and abstract are appropriate for the contents of the text.

Introduction: The authors summarized the current available information on this topic in a clear and concise manner. Although the authors explain the advantages of using a clinical data warehouse for overcoming certain limitations of retrospective data collection, the statement that using a CDW approach overcomes the issue of "retrospective studies having an inferior level of evidence compared with prospective studies" is not entirely adequate and should be revised.

Materials and methods: The patients appear to represent the whole experience of the investigators. The methodology for patient inclusion and exclusion was presented clearly. The methodology for clinical and pathological definitions is explained but the interpretation of one of the key items in this study, namely the definition of Ki67 categories is not mentioned. Perhaps elaborating on the cut-off values for inclusion into the categories of Ki67 1+, 2+ and 3+ would increase the readers' understanding of the subject.

The statistical analysis is described, however, it is unclear whether the normality of the distribution of data was tested (for example, if a Shapiro-Wilk test was performed).

Results: The authors adequately presented their findings. The information presented is nicely supported by the figure and tables. In table 4, the definition of "Ki67>4" warrants further explanation as to its meaning, as previously mentioned.

Discussions: The results are discussed in relation to the evidence currently available in the literature. The limitations and strengths of the present study are adequately presented.

Conclusions: The conclusions of the authors are appropriately cautious given the limitations of the study.

Lastly, while the use of language is mostly sound, a revision of grammar and syntax is required in order to address any minor errors which were identified, thus improving the flow and readability of the text.

Author Response

Thank you for your valuable comments.

The manuscript submitted by Kim et al. analyzed the clinicopathological characteristics and treatment outcomes of patients diagnosed with refractory metastatic triple negative breast cancer (mTNBC) in relation to a cohort of patients diagnosed with metastatic triple-negative breast cancer which responded to adjuvant treatment. To this end, the authors conducted a retrospective observational study using a clinical data warehouse (CDW) approach, comprising 415 patients which presented with mTNBC between January 1997 and December 2019 in a single institution, which were divided into refractory and non-refractory cohorts based on their response to neoadjuvant/adjuvant chemotherapy and compared in terms of clinical data, pathological data, laboratory data and treatment regimens, as well as disease-free survival and overall survival. The authors concluded that patients with refractory mTNBC were younger, had more aggressive tumor biology and significantly poorer outcomes in terms of overall survival, disease-free survival and progression-free survival compared to the non-refractory cohort. The main strength of this study is that it addresses a relevant research question, with significant implications for clinical practice.

Title and abstract: The title and abstract are appropriate for the contents of the text.

Introduction: The authors summarized the current available information on this topic in a clear and concise manner. Although the authors explain the advantages of using a clinical data warehouse for overcoming certain limitations of retrospective data collection, the statement that using a CDW approach overcomes the issue of "retrospective studies having an inferior level of evidence compared with prospective studies" is not entirely adequate and should be revised

Thank you for your comment. We thought the expression is inadequate so removed the expression “retrospective studies having an inferior level of evidence compared with prospective studies”.

Materials and methods: The patients appear to represent the whole experience of the investigators. The methodology for patient inclusion and exclusion was presented clearly. The methodology for clinical and pathological definitions is explained but the interpretation of one of the key items in this study, namely the definition of Ki67 categories is not mentioned. Perhaps elaborating on the cut-off values for inclusion into the categories of Ki67 1+, 2+ and 3+ would increase the readers' understanding of the subject.

Thank you for your comment. We added the definition of Ki-67 1+ ≤25%, 2+ (26-50%) 3+ 51-75%, 4+ 76-100% and added the sentence as follows.

Ki-67 index is ranged measured by the percentage of the positive stained tumor nuclear cells out of all tumor cells. It ranged from 0% to 100% and we defined 1+ (≤25%), 2+ (25-50%), 3+ (51-75%), and 4+ (76-100%).

The statistical analysis is described, however, it is unclear whether the normality of the distribution of data was tested (for example, if a Shapiro-Wilk test was performed).

Thank you for your comment. We added the sentence as below.

The Chi-square test was used to compare the distribution of categorical variables.

Results: The authors adequately presented their findings. The information presented is nicely supported by the figure and tables. In table 4, the definition of "Ki67>4" warrants further explanation as to its meaning, as previously mentioned.

Thank you for your comment. There was an error in writing Ki-67 4+ as Ki-67 ≥4. Thanks to you, we were able to correct the error.

Table 4. Univariable and multivariable analyses of risk factors of overall survival.

Characteristics                  

Univariable analysis

Multivariable analysis

HR (95% CI)

P-value

HR (95% CI)

P-value

Age < 40

0.997

(0.793–1.252)

0.977

NA

NA

Non-IDC histology

0.677

(0.403–1.139)

0.141

NA

NA

Ki-67 4+

0.978

(0.795–1.201)

0.829

NA

NA

Bone metastasis

1.299

(1.045–1.615)

0.018

1.184

(0.649- 1.77)

0.135

Brain metastasis

1.162

(0.820–1.646)

0.398

NA

NA

Liver metastasis

1.947

(1.506–2.518)

< 0.001

2.009

(1.552–2.600)

< 0.001

Lung metastasis

0.918

(0.746–1.129)

0.416

NA

Leptomeningeal seeding

1.744

(1.230–2.473)

0.002

1.862

(1.311–2.645)

0.001

Discussions: The results are discussed in relation to the evidence currently available in the literature. The limitations and strengths of the present study are adequately presented.

Thank you for your comment.

Conclusions: The conclusions of the authors are appropriately cautious given the limitations of the study.

Lastly, while the use of language is mostly sound, a revision of grammar and syntax is required in order to address any minor errors which were identified, thus improving the flow and readability of the text.

Thank you for your comment. We fixed some grammar problems.

Reviewer 3 Report

The authors presented a very interesting issue on Triple Negative Breast Cancer.

I only have two little tips: 

61 a full stop is missing at the end of the sentence.

200 I would transfer the beginning of the table from page 6 to page 7.

Author Response

Thank you for your valuable comments.

The authors presented a very interesting issue on Triple Negative Breast Cancer.

I only have two little tips: 

61 a full stop is missing at the end of the sentence.

200 I would transfer the beginning of the table from page 6 to page 7.

Thank you for your comment. We added a full stop line 61 and transferred the table to page 7.

Reviewer 4 Report

In this manuscript, the authors conduct an analysis of metastatic TNBC patients from a patient registry or clinical data warehouse. They describe the patient and tumor level characteristics that make up the cohort of mTNBC patients and stratify them by refractory, non-refractory patients. The Refractory patients were then further stratified by those that were refractory after surgery and those that progressed during neo-adjuvant chemotherapy. The authors provide relevant clinical endpoint estimates (DFS, PFS, 2nd PFS, OS) and a description of risk factors that impact overall survival. 

Overall, this manuscript is well written and contributes to the growing field of real world data using patient registries. The authors elegantly describe how despite the limitations of this data source, the clinical endpoints measured herein are from populations of patients who may be generally excluded from clinical trials therefore literature estimates are not applicable to these patients. Furthermore, they provide an assessment of risk factors that give further insights into the differences in outcomes amongst mTNBC patients. These data emphasize the importance of the collection and curation of databases that will inform clinicians and researchers. I recommend acceptance of this manuscript with a minor revision.

Minor revision:

I recommend the authors to reformat Table 1 so that the top header fits with at least 50% of the lower table. Some restructuring of the table would clarify the content of the columns. 

Author Response

Thank you for your valuable comments.

In this manuscript, the authors conduct an analysis of metastatic TNBC patients from a patient registry or clinical data warehouse. They describe the patient and tumor level characteristics that make up the cohort of mTNBC patients and stratify them by refractory, non-refractory patients. The Refractory patients were then further stratified by those that were refractory after surgery and those that progressed during neo-adjuvant chemotherapy. The authors provide relevant clinical endpoint estimates (DFS, PFS, 2nd PFS, OS) and a description of risk factors that impact overall survival. 

Overall, this manuscript is well written and contributes to the growing field of real world data using patient registries. The authors elegantly describe how despite the limitations of this data source, the clinical endpoints measured herein are from populations of patients who may be generally excluded from clinical trials therefore literature estimates are not applicable to these patients. Furthermore, they provide an assessment of risk factors that give further insights into the differences in outcomes amongst mTNBC patients. These data emphasize the importance of the collection and curation of databases that will inform clinicians and researchers. I recommend acceptance of this manuscript with a minor revision.

Minor revision:

I recommend the authors to reformat Table 1 so that the top header fits with at least 50% of the lower table. Some restructuring of the table would clarify the content of the columns. 

Thank you for your comment. We reformatted Table 1

Recurrence of mTNBC

after curative surgery

De novo mTNBC

Nonrefractory

mTNBC

     Refractory mTNBC

Refractory mTNBC

after surgery

Unresectable mTNBC

d/t progression

during NAC

Patient, no.

69

131

207

44­­

251

Age

At diagnosis,

median, yr. (range)

49 (28–89)

51 (29–80)

45 (24–90)

46 (28–76)

46 (24–90)

< 40

15 (22)

20 (15)

67 (32)

16 (36)

83 (33)

< 60

53 (77)

111 (85)

192 (93)

41(93)

233 (93)

≥ 60

16 (23)

20 (15)

15(7)

3 (7)

18 (7)

Menopausal status at diagnosis

Premenopause

29 (42)

52 (40)

130 (63)

27 (61)

157 (63)

Postmenopause

34 (49)

64 (49)

64 (31)

14 (32)

78 (31)

Unknown

6 (9)

15 (11)

13 (6)

3 (7)

16 (6)

Histologic subtypes

Ductal

62 (90)

113 (86)

180 (87)

43 (98)

223 (89)

Lobular

1 (1)

3 (2)

2 (1)

0

2 (1)

Metaplastic

1 (1)

9 (7)

17 (8)

0

17 (7)

Other

5 (7)

6 (5)

8 (4)

1 (2)

9 (4)

Ki-67

1+

6 (9)

29 (22)

19 (9)

6 (14)

25 (10)

2+

12 (17)

25 (19)

38 (18)

7 (16)

45 (18)

≥ 3

46 (67)

62 (47)

146 (71)

31 (70)

177 (71)

Unknown

5 (7)

15 (11)

3 (1)

0

3 (1)

BRCA mutation

BRCA1

3/13 (23)

2/15 (13)

9/53 (17)

1/5 (20)

10/58 (17)

BRCA2

1/13 (8)

0/15

3/53 (6)

0/5

3/58 (5)

Distant metastasis site at diagnosis of MBC

 Bone

29 (42)

36 (27)

51 (25)

11 (25)

62 (25)

 Brain

0

8 (6)

31 (15)

4 (9)

35 (14)

 Liver

16 (23)

20(15)

34 (16)

6 (14)

40 (16)

 Lung

24 (35)

59 (45)

73 (35)

18 (41)

91 (36)

First-line palliative chemotherapy

AC

35 (51)

5 (4)

7 (3)

4 (9)

Taxane only

9 (13)

38 (29)

43 (21)

6 (14)

49 (20)

Taxane + G

0

15 (11)

8 (4)

1 (2)

Taxane + platinum

10 (15)

34 (26)

51 (25)

13 (30)

64 (25)

GP (C)

3 (4)

6 (5)

60 (29)

11 (25)

71 (28)

NX

0

0

2 (1)

1 (2)

Capecitabine

8 (12)

25 (19)

18 (9)

7 (16)

Other

4 (6)

8 (6)

18 (9)

1 (2)

Second-line palliative chemotherapy

Patients, no.

57/69 (83)

105/131 (80)

167/207 (81)

35/44 (80)

202 (80)

AC

16 (28)

10 (10)

8 (5)

1 (3)

Taxane only

12 (21)

7 (7)

24 (14)

3 (9)

Taxane + G

0

1 (1)

0

0

Taxane + platinum

11 (19)

12 (11)

19 (11)

2 (6)

GP (C)

5 (9)

27 (26)

43 (26)

8 (23)

51 (25)

NX

1 (2)

0

4 (2)

1 (3)

Capecitabine

10 (18)

39 (37)

32 (19)

13 (37)

45 (22)

Other

2 (4)

9 (9)

37 (22)

7 (20)

Abbreviations: mTNBC, metastatic triple-negative breast cancer; AC, adriamycin/cyclophosphamide; G, gemcitabine; GP (C), gemcitabine/cisplatin (carboplatin); NX, navelbin/xeloda.

Round 2

Reviewer 1 Report

I would like to thank the authors for their efforts regarding my comments. This study is interesting, however, major limitations due to the design are still not enough discussed.

“the data of patients’ characteristics in supplement 1 seems to be important for predicting the patient’s prognosis. “: Thank you for adding these data. However, I don’t see comorbidities, lab results, … whereas you talk about it in your materials & methods section. Authors have to described in their material & methods only data they have used for their analysis.

Please add clearly in the discussion that these missing data could be major confounding factors for your conclusions. It must be added in the discussion.

“How do you identify TNBC patients before the characterization of HER2 status” – I would like to ask: from which year was it tested in Korea? FDA approved the test in 1998, and your study began in januaray 1997. How are you sure that first cases were TNBC and not only HR negative ?

I would like authors add a comment on brain imagery: is it performed systematically or not, only on symptoms, do they have the information or not. This is a major point. There is nothing about that in the paper, whereas authors described brain metastases.

“date of disease progression or start of second line chemotherapy are very different endpoint. They should not be included in the same endpoint. » : « However, such cases are seemed to be very few and will not significantly affect the analysis results.” : If you can not identified these cases, it should be add in the discussion.

Your response” patients who did not receive palliative chemotherapy due to poor general condition may also be excluded from the patient group.” have to be add in the discussion.

Regarding my comment : “A huge quantity of information are missing, and they have to be discussed in this section. Clinical parameters (PS, co-morbidities, …), stage of disease, imagery at baseline, …. All these missing data obviously biased conclusions.

Your response:” Thank you for your comment. It was not possible to include all necessary clinical parameters in this paper.”

I agree, but you have to highlight these limits in your discussion very clearly.

Author Response

the data of patients’ characteristics in supplement 1 seems to be important for predicting the patient’s prognosis. “: Thank you for adding these data. However, I don’t see comorbidities, lab results, … whereas you talk about it in your materials & methods section. Authors have to described in their material & methods only data they have used for their analysis.

Please add clearly in the discussion that these missing data could be major confounding factors for your conclusions. It must be added in the discussion.

Thank you for your comment. First of all, I am very surprised and thankful that your comments are very valuable and important. We added that content in the discussion part.

We could not analyze all factors that might affect the patients’ prognosis, such as clinical stages, comorbidities, and other blood lab results. It is possible that these variables could affect as confounding factors.

“How do you identify TNBC patients before the characterization of HER2 status” – I would like to ask: from which year was it tested in Korea? FDA approved the test in 1998, and your study began in January 1997. How are you sure that first cases were TNBC and not only HR negative ?

Thank you for the comment. You pointed out a very important thing. Reviewing our data, the first HER2 test result appeared has been conducted since 2003 in SMC. When we extracted the patient group, the search condition was set from patients included in the breast cancer registry since 1997. Patients who had been diagnosed before 2003 and who relapsed and tested HER2 status after that were included. The patients were enrolled in the breast cancer registry prior to the introduction of the HER2 test, thus they were extracted into the patient group under the search criteria we have set. Therefore, the patient group has all HER2 test results performed after 2003. Five of these patients belonged, all diagnosed before 2000, relapsed after 2003 and classified as TNBC with HER2 test result. These results were derived from using 1997 as the patient search condition. If this is a problem, the text, 1997 in this study may be revised to 2003. However, we used the CDW method to reduce the human error that occurs when a manual search for data through chart review, and we wanted to explain how to extracted data, the process, in the paper.

I would like authors add a comment on brain imagery: is it performed systematically or not, only on symptoms, do they have the information or not. This is a major point. There is nothing about that in the paper, whereas authors described brain metastases.

 Thank you for your comment. We added the sentence as followed.

Brain imaging was not performed at regular intervals, but performed following a clinician’s decision when the symptoms of brain metastasis occurred. Hence the brain metastasis might differ from the actual occurrence of brain metastasis.

 “date of disease progression or start of second line chemotherapy are very different endpoint. They should not be included in the same endpoint. » : « However, such cases are seemed to be very few and will not significantly affect the analysis results.” : If you can not identified these cases, it should be add in the discussion.

Thank you for your comment. We added the sentence as followed.

In addition, defined PFS period for data extraction, beginning of the first line chemotherapy until the start of second line chemotherapy is generally consistent but might differ from correct PFS period.

Your response” patients who did not receive palliative chemotherapy due to poor general condition may also be excluded from the patient group.” have to be add in the discussion.

 Thank you for your comment. We added the sentence about you mentioned as followed.

Patients who did not received systemic chemotherapy due to poor general condition were not included in this cohort, also.

Regarding my comment : “A huge quantity of information are missing, and they have to be discussed in this section. Clinical parameters (PS, co-morbidities, …), stage of disease, imagery at baseline, …. All these missing data obviously biased conclusions.

Your response:” Thank you for your comment. It was not possible to include all necessary clinical parameters in this paper.”

I agree, but you have to highlight these limits in your discussion very clearly.

Thank you for your comment. We added the sentence as followed.

We could not analyze all factors that might affect the patients’ prognosis, such as clinical stages, comorbidities, and other blood lab results. It is possible that these variables could affect as confounding factors.

Revised discussion  

Our study had several limitations. This study was a retrospective analysis, and the study cohort was derived from a single large center in Korea. As a tertiary hospital, our patient cohort might not represent other institutions’ metastatic TNBC patients. Also, our patient population was of a single ethnicity, and its analysis might have yielded results different from those of other ethnic groups. Brain imaging was not performed at regular intervals, but performed following a clinician’s decision while the symptoms of brain metastasis occurred. Hence the brain metastasis might differ from the actual occurrence of brain metastasis. In addition, defined PFS period for data extraction, beginning of the first line chemotherapy until the start of second line chemotherapy is generally consistent but might differ from correct PFS period. We could not analyze all factors that might affect the patients’ prognosis, such as clinical stages, comorbidities, and other blood lab results. It is possible that these variables could affect as confounding factors.

However, in this study, we performed patient selection and data collection using a real-time site-specific cancer registry integrated into a CDW approach, and these efforts could have reduced human errors. Furthermore, the research approach suggested by the study using a preprocessed dataset with data quality measures can help accelerate the generation of real-world evidence by facilitating the cycle of hypothesis derivation and verification. Another limitation is that patients with a history of systemic chemotherapy were selected during the development of the initial study cohort. Therefore, it is possible that patients who only received local treatment such as intrathecal methotrexate or whole-brain radiation therapy but not systemic chemotherapy were excluded from the baseline cohort. Patients who did not received systemic chemotherapy due to poor general condition were not included in this cohort, also. For this reason, it is estimated that the distant recurrence rate of 12.8% (451 of 3531) of TNBC shown in this paper seemed to be lower than the distant recurrence rates of around 20% presented in other articles

Round 3

Reviewer 1 Report

I would like to thank the authors for their responses and corrections.